# Dual-Gas Sensor of CH_4_/C_2_H_6_ Based on Wavelength Modulation Spectroscopy Coupled to a Home-Made Compact Dense-Pattern Multipass Cell

**DOI:** 10.3390/s19040820

**Published:** 2019-02-17

**Authors:** Xing Tian, Yuan Cao, Jiajin Chen, Kun Liu, Guishi Wang, Tu Tan, Jiaoxu Mei, Weidong Chen, Xiaoming Gao

**Affiliations:** 1Anhui Institute of Optics and Fine Mechanics, Chinese Academy of Science, Hefei 230031, China; txtian@mail.ustc.edu.cn (X.T.); cy0413@mail.ustc.edu.cn (Y.C.); chenjiajin64@163.com (J.C.); liukun@aiofm.ac.cn (K.L.); gswang@aiofm.ac.cn (G.W.); tantu@aiofm.ac.cn (T.T.); jxmei@aiofm.ac.cn (J.M.); 2Science Island Branch of Graduate School, University of Science and Technology of China, Hefei 230026, China; 3Laboratoire de Physicochimie de l’Atmosphère, Université du Littoral Côte d’Opale, 189A, Av. Maurice Schumann, 59140 Dunkerque, France; chen@univ-littoral.fr

**Keywords:** gas sensor, methane, ethane, wavelength modulation spectroscopy, multipass absorption cell

## Abstract

A sensitive dual-gas sensor for the detection of CH_4_ and C_2_H_6_ is demonstrated. Two tunable semiconductor lasers operating at 1.653 μm (for CH_4_ monitoring) and 1.684 μm (for C_2_H_6_) were used as the light source for spectroscopic measurements of CH_4_ and C_2_H_6_. Long-path absorption in a home-made compact dense-pattern multipass cell (L_eff_ = 29.37 m) was employed, combined with wavelength modulation and second harmonic detection. Simultaneous detection of CH_4_ and C_2_H_6_ was achieved by separated wavelength modulations of the two lasers. Modulation frequencies and amplitudes were optimized for sensitivity detection of CH_4_ and C_2_H_6_ simultaneously. The dual-gas sensor exhibits 1σ detection limits of 1.5 ppbv for CH_4_ in 140 s averaging time and 100 ppbv for C_2_H_6_ in 200 s.

## 1. Introduction

Natural gas is the third largest source of thermal energy in the world, accounting for 22.1% of primary energy consumption after petroleum (31.9%) and coal (27.1%). The methane (CH_4_) content in natural gas reaches over 90%. Biogas, which contains mainly CH_4_, is a naturally occurring gas that forms from the breakdown of organic matter in the presence of anaerobic bacteria. CH_4_ is also a main greenhouse gas, whose greenhouse effect is 25 times higher than that of carbon dioxide (CO_2_). Ethane (C_2_H_6_) is the second largest component in natural gas, accounting for about 5–10%. It can be used as an indicator to discriminate whether the measured leaking gas is from natural gas, underground biogas, or other combustible gas. Monitoring C_2_H_6_ can be applied to diagnose leaking natural gas pipelines [1,2,3,4,5,6]. 

Laser spectroscopy-based gas sensors is an effective tool for environmental monitoring, medical diagnosis, industrial process control, food industry, and pollution monitoring [7], and includes quartz-enhanced photoacoustic spectroscopy (QEPAS) [8,9,10], off-axis integrated cavity output spectroscopy (OA-ICOS) [11], and tunable diode laser absorption spectroscopy (TDLAS) with a multipass cell. In general, long-path absorption is combined with wavelength modulation to improve measurement sensitivity and precision [12,13]. Krzempek demonstrated C_2_H_6_ concentration measurement based on 2f wavelength modulation spectroscopy (WMS) using a continuous-wave (CW) distributed feedback diode laser emitting in the mid-infrared near 3.36 µm. A 1σ minimum detectable concentration of 740 pptv with a 1 s lock-in amplifier time constant was achieved [14]. Zheng and co-workers utilized two CW interband cascade lasers (ICLs) operating at 3291 nm (CH_4_) and 3337 nm (C_2_H_6_) for the simultaneous measurement of CH_4_/C_2_H_6_ in the mid-infrared. Detection limits of ~1.7 ppbv for CH_4_ in 9 s and ~0.36 ppbv for C_2_H_6_ in 65 s [15] were achieved. Li et al. realized ppbv-level measurement of C_2_H_6_ using 3.34 µm CW-ICL-based WMS with a detection limit of ~1.2 ppbv in a 4 s averaging time [16].

Though C_2_H_6_ absorption intensity is relatively weak in the near-infrared spectral region [17,18], the photonic devices in this spectral region have the characteristics of being low cost and relatively mature. Therefore, it is of great significance to develop a high-sensitivity dual-gas sensor for CH_4_/C_2_H_6_ by TDLAS for leakage detection applications [19].

In this paper, we report on the development of an optical sensor for the simultaneous measurement of CH_4_ (at 1.653 µm) and C_2_H_6_ (at 1.684 µm) based on long-path absorption in a newly home-made dense-pattern multipass cell combined with wavelength modulation and second harmonic detection [20]. 

## 2. Basics of Wavelength-Modulation Spectroscopy

Wavelength-modulation spectroscopy (WMS) is a spectroscopic measurement technique widely used for trace gas measurements in harsh environments owing to its high noise-rejection capability over direct absorption spectroscopy. Only a brief description of some of the fundamental theory will be given here to explain certain experimental choices (e.g., modulation index) made in this study [21,22,23]. The instantaneous laser frequency with sinusoidal modulation can be written as:
(1)$$\nu (t)=\nu_{0}+\Delta v\cos(\omega t)$$
where *ν*_0_ (cm^−1^) is the center laser emission frequency, *ω* is the angular frequency, and Δ*ν* (cm^−1^) is the modulation depth.

The time-dependent transmission coefficient *τ*(*ν*) can be expanded in a Fourier cosine series:
(2)$$\tau [\nu (t)]=\tau [\nu_{0}+\Delta v\cos(\omega t)]=\sum_{n=0}^{n=+\infty }H_{n}(\nu_{0},\Delta v)\cos(n\omega t)$$
where *H_n_* (*ν*_0_, Δ*ν*) is the n-th component of the Fourier series, which can be expressed as:
(3)$$H_{0}(\nu_{0},\Delta v)=\frac{1}{2\pi }\int_{-\pi }^{\pi }\tau (\nu_{0}+\Delta v\cos\theta )d\theta \;(n=0)$$
(4)$$H_{n}(\nu_{0},\Delta v)=\frac{1}{\pi }\int_{-\pi }^{\pi }\tau (\nu_{0}+\Delta v\cos\theta )\cos(n\theta )d\theta \;(n>0)$$


In practice, second harmonic is usually used for the detection (n = 2):
(5)$$H_{2}(\nu_{0},\Delta v)=\frac{1}{\pi }\int_{-\pi }^{\pi }\tau (\nu_{0}+\Delta v\cos\theta )\cos(2\theta )d\theta \;(n=2)$$


In the case of trace gas absorption described by the Beer-Lambert law [24] (*α*(*ν*)*L* << 1 ), the transmission can be simplified as:
(6)$$\tau (\nu )=\frac{I_{out}(\nu )}{I_{in}(\nu )}=\exp[-\alpha (\nu )L]\approx 1-\alpha (v)L=1-N_{0}SCL\chi (v-v_{0})$$
where *I_out_(ν)* and *I_in_(ν)* are the transmitted and incident light intensities at frequency *ν*, respectively, L is the optical absorption path length in [cm]. *α*(*ν*) = *σ(ν*)*N*_0_*C* = *χ*(*ν* − *ν*_0_)*SN*_0_*C* (at standard temperature and pressure) is the absorption coefficient of the target gas with concentration *C* (mixing ratio) and the Loschmidt constant *N*_0_ = 2.6868 × 10^19^ molecule/cm^3^. *σ*(*ν*) = *χ*(*ν* − *ν*_0_)*S* is the absorption cross-section of the target gas with *S* the line strength in [cm^−1^/(molecules/cm^−2^)] and *χ*(*ν* − *ν*_0_) the absorption line shape function. *ν*_0_ is the central frequency of the absorption line.

From Equations (5) and (6), the Fourier series of the second harmonic can be expressed as:
(7)$$H_{2}(\nu_{0},\Delta v)=\frac{-N_{0}SCL}{\pi }\int_{-\pi }^{\pi }\chi (\nu_{0}+\Delta v\cos\theta )\cos(2\theta )d\theta$$


According to Equation (7), the harmonic signal is hence proportional to the concentration of the gas. In addition to the absorption parameters, the WMS-2f signal also depends on the modulation depth Δν. The WMS-2f signal can be maximized by choosing the optimal modulation index m, which is defined as:m = Δν/Δν_D_(8)
where Δν_D_ [cm^−1^] is the full width at half maximum (FWHM) of the absorption line shape, and the 2f signal amplitude is maximized at a modulation index m = 2.2 [25,26]. 

In addition, the background signal is caused by the 2f residual–amplitude–modulation (2f-RAM) signal in the presence of linear intensity modulation [27]. In the present work, we use the second harmonic signal deducting the background to determine the concentrations of CH_4_ and C_2_H_6_.

## 3. Experimental Details 

### 3.1. Design of the Dual-Gas Sensor for CH_4_ and C_2_H_6_

The developed dual-gas sensor is shown in Figure 1. Two tunable distributed feedback (DFB) diode lasers operating at 1.653 µm and 1.684 µm were used for the measurements of CH_4_ and C_2_H_6_, respectively. The temperature and current of both lasers were controlled by laser diode controllers LDC501 (Stanford Research Inc., Sunnyvale, CA, USA). Both lasers currents were scanned by feeding an external voltage ramp at a rate of 1 Hz from a function generator (RIGOL DG4162). Wavelength modulations (6 kHz for CH_4_ and 0.6 kHz for C_2_H_6_) were achieved through the modulation of each laser current via a sine form wave from lock-in amplifiers #1 and #2 (SR 830 DSP, Stanford Research Systems), respectively. The voltage ramp and the sine wave were combined with a home-made adder before being coupled to the laser diode controllers.

The two laser beams were coupled together into a 2 × 1 single mode standard coupler (DWBC) with the maximum insertion loss 0.3 dB/km. The output beams were collimated through a single fiber optical collimator with the beam diameter not exceeding 0.35 mm and then injected into a home-made multipass absorption cell. The multipass cell consisted of two 2” silver-coated concave spherical mirrors separated by a distance of 12 cm, which was similar to that previously reported in [20]. An effective optical path of 29.37 m was achieved with 243 multiple reflections of laser beam inside the cell. 

The transmitted beam was detected with an indium gallium arsenic photodetector (PDA20CS-ES) via a focusing lens (f = 30 mm) as shown in Figure 2. The signal detected by the photodetector was sent to lock-in amplifiers #1 and #2, respectively, for demodulations at 6 kHz for CH_4_ and 0.6 kHz for C_2_H_6_, so as to distinguish the signals from different lasers and realize simultaneous detection of CH_4_ and C_2_H_6_. The demodulated signals were subsequently digitalized with a data acquisition card (DA, NI-USB-6212) and displayed on a laptop via LabVIEW interface.

### 3.2. Spectral Line Selection

C_2_H_6_ shows an unresolvable absorption band near 1.684 µm [28,29]. There is no information on its spectral line parameters in the common database (such as HITRAN) for ethane in the near-infrared. In order to determine the laser temperature and the current corresponding to the maximum ethane absorption, 10,000 ppm ethane gas was filled in the multipass cell at a pressure of 0.5 atm. Direct C_2_H_6_ absorption spectrum was obtained by a wavelength scan of the 1.684 µm DFB laser across the C_2_H_6_ absorption lines, as shown in Figure 3. The laser temperature and the current needed to probe the strong absorption band of C_2_H_6_ near 1.684 µm were found to be 31 °C and 78 mA, respectively. 

While the best CH_4_ absorption lines within the spectral tuning range of the used 1.653 µm diode laser are the R_3_ triplet of the 2ν_3_ band near ~6046.95 cm^−1^, they have the highest line intensities and are free of interference from other molecules (such as H_2_O and CO_2_). The R_3_ triplet of the 2ν_3_ band at ~6046.95 cm^−1^ was selected for the measurement of CH_4_ concentration. Figure 4 shows an absorption spectrum of 492 ppm CH_4_ at a pressure of 0.5 atm. The laser center current was set at 85 mA with a temperature set at 22 °C for targeting the selected CH_4_ absorption lines. The amplitude of the triangular ramp was 1.3 V, which scanned the laser diode current from 52.5 to 117.5 mA (50 mA/V).

The calibration curves of laser current vs. frequency were realized with the help of a wavelength meter (Bristol 621).

## 4. Results and Discussions

### 4.1. Optimization of Modulation Amplitude for WMS

In WMS, the second harmonic (2f) signal amplitude depends on the modulation amplitude [30]. In order to obtain optimal modulation amplitude for maximum second harmonic signals, a mixture of 200 ppm CH_4_ and 300 ppm C_2_H_6_ in nitrogen was prepared using a gas diluter (Environics SERIES 4000) and then filled in the multipass cell at 1 atm. The 2f signals of 200 ppm CH_4_ and 300 ppm C_2_H_6_ at different modulation amplitudes were investigated. The measured results of the 2f signals versus the modulation amplitudes are shown in Figure 5. As can be seen, the optimal modulation amplitude was 0.25 V for CH_4_ and 0.36 V for C_2_H_6_. The optimal modulation amplitude of the sine wave for CH_4_ detection was found to be 0.25 V, and the corresponding laser wavelength modulation amplitude was Δν_m_ ≈ 0.2921 cm^−1^. According to the HITRAN database, the 200 ppm CH_4_ line width (HWHM) of the targeted absorption lines was Δν_D_ ≈ 0.1316 cm^−1^. The optimal modulation amplitude Δν_m_ ≈ 2.22 Δν_D_ for wavelength modulation with second harmonic demodulation was basically consistent with the theoretical result Δν_m_ = 2.2 Δν_D_ [25,26]. 

### 4.2. Optimization of Modulation Frequency and Phase for WMS

In addition to the modulation amplitude, the modulation frequency and phase may also affect the amplitude of the second harmonic signal [31,32]. Under the condition of the optimal modulation amplitude, the effects of the modulation frequency and phase on the second harmonic signal amplitude were investigated, experimentally. Figure 6 shows the phase effect and Figure 7 shows the modulation frequency effect. The optimal phase was found to be −140° for both CH_4_ and C_2_H_6_ detections. The optimal modulation frequency was 6 kHz for CH_4_ detection and 0.6 kHz for C_2_H_6_ detection.

### 4.3. Investigation on the Potential Cross-Talk 

Two lasers operating at different wavelengths were injected into the same absorption cell. It might cause cross-talk effects between two signals if the selection of the modulation frequencies is not appropriate [33]. For example, cross-talk occurs when modulation frequencies are fixed at 6 kHz for CH_4_ and at 5.8 kHz for C_2_H_6_ detection, as shown in Figure 8. The cross-talk effect was determined by the low-pass filter of the lock-in amplifier. The amplitude of the sinuous residual was determined by the stopband edge and the response of the filter. 

Figure 9 shows the experimental study of the cross-talk effects in the simultaneous detection of CH_4_ (190 ppmv) and C_2_H_6_ (170 ppmv). In order to verify that there is no cross-talk between two signal channels, two lasers work simultaneously under wavelength modulation at their specific optimal modulation frequencies, i.e., 6 kHz for CH_4_ and 0.6 kHz for C_2_H_6_. Figure 9a plots the second harmonic signals of CH_4_ with and without the operation of the laser for C_2_H_6_ detection. The amplitude of the CH_4_ signal was 7.931 mV when the two lasers were on. When the C_2_H_6_ laser was turned off, the amplitude of the CH_4_ signal was 7.926 mV. The difference between the peak heights of the CH_4_ second harmonic signals was about 0.06%. Figure 9b compares the case for C_2_H_6_ detection. The amplitude of the C_2_H_6_ signal was 4.873 mV when the two lasers worked together. When the CH_4_ laser was turned off, the amplitude of the C_2_H_6_ signal was 4.865 mV. The deviation between the peak heights of C_2_H_6_ second harmonic signals was 0.16%. It can be concluded that the cross-talk effects can be ignored when the two lasers are both operational at their specific optimal modulation frequency in the present work. 

The optimal parameters used for the present work are listed in Table 1 and are the default values in the subsequent experimental measurements.

### 4.4. Baseline Removal

Using WMS could significantly reduce the 1/f noise. However, when the baseline is not a “linear” ramp, the derivation cannot remove such “curve shape” baseline [27]. In the present work, the baseline of the sensor was investigated using high purity (99.99%) nitrogen (N_2_) filled in the multipass cell. The N_2_ background baselines were measured in the CH_4_ and C_2_H_6_ channels as shown in Figure 10a,b, respectively. In the present work, the 2f signals of CH_4_ and C_2_H_6_ were obtained by subtracting the N_2_ background baselines. As an example, Figure 11a shows a 0.6 ppm CH_4_ signal, obtained directly from the lock-in amplifier associated with the baseline. Figure 11b shows the 2f signal of 0.6 ppmv CH_4_ after subtracting the baseline shown in Figure 10a.

The standard reference gases of 492 ppmv CH_4_ and 500 ppmv C_2_H_6_ were mixed with high purity nitrogen to obtain a gas mixture with concentrations ranging from 10 ppmv to 100 ppmv. A gas dilution system (Model 4000, Environics Inc., Tolland, CT, USA) was used to make these mixtures. The second harmonic signals of CH_4_ and C_2_H_6_ at different concentrations and under normal atmospheric pressure are plotted in Figure 12a,b.

Based on the results shown in Figure 12a,b, the second harmonic amplitudes vs. different concentrations are plotted in Figure 13a,b for CH_4_ and C_2_H_6_, respectively. Good linear correlations are observed with a correlation coefficient of 99.97% for CH_4_ and 99.98% for C_2_H_6_, which can be used for calibration of the measured 2f signal amplitudes to their corresponding concentrations. 

### 4.5. Allan Deviation

In order to determine the sensor stability and limit of detection (LoD), analyses of Allan variance were carried out. Figure 14a,b plot time series measurements of CH_4_ and C_2_H_6_ at a constant concentration of 2 ppmv and 32 ppmv diluted with the concentration 2.1 ppmv of CH_4_ and 500 ppmv of C_2_H_6_ standard gas, respectively. The corresponding Allan deviations as a function of the averaging time t are plotted in Figure 14c,d. Allan variance analysis shows that the measurement precision of the developed dual-gas sensor was ~20 ppbv for CH_4_ and 1.2 ppmv for C_2_H_6_ with an acquisition time of 1 s. A detection limit of 1.5 ppbv for CH_4_ could be achieved by averaging in 140 s and 100 ppbv could be achieved for C_2_H_6_ by averaging 200 s. 

### 4.6. Simultaneous Detection of Methane and Ethane Leakage

The sensor performance was evaluated by simultaneous detection of methane and ethane leakage with the current locked at the absorption peak, respectively. The outside air was pumped into an airbag (60 cm × 80 cm) and then injected into the gas cell through a long sampling tube.

Based on the calibration curves shown in Figure 13, measured second signals of CH_4_ and C_2_H_6_ can be converted to their corresponding concentrations. 

Time series measurements of the test are plotted in Figure 15. Methane and ethane in air were measured in the period of t_1_. The atmospheric CH_4_ concentration of about 2.1 ppmv was observed, while the C_2_H_6_ concentration in ambient air was lower than the detection limit of the current sensor and was not observed. CH_4_ and C_2_H_6_ were then added in the airbag and changes in the concentrations of CH_4_ and C_2_H_6_ were observed in the period of t_2_. The CH_4_ concentration varied from 2.1 to 14.1 ppmv and the C_2_H_6_ concentration varied from 0 to 344.7 ppmv. Subsequently, we stopped the “leakage” of CH_4_ / C_2_H_6_ and flushed only air into the absorption cell; the concentrations of CH_4_ and C_2_H_6_ returned to their atmospheric levels (t_3_). 

It is worth noting that the minimum detection limit of the commercial PGC ethane analyzer (Schutz, German) is 10 ppmv. The present experimental results of simultaneous detection of methane and ethane demonstrate the real potential of the developed dual-gas sensor of CH_4_/C_2_H_6_ for use in leakage detection.

## 5. Conclusions and Outlook

In this paper, we demonstrated a compact dual-gas sensor for monitoring CH_4_ and C_2_H_6_ based on wavelength modulation spectroscopy coupled to a compact home-made dense-pattern multipass cell using a single photodetector. Detection limits were 1.5 ppbv for CH_4_ by averaging 140 s and 100 ppbv for C_2_H_6_ by averaging 200 s. The developed dual-gas sensor can be used to distinguish between natural gas and biogas, which has strong potential for gas sensing applications in natural gas pipeline leakage analysis. 

## Figures and Tables

**Figure 1 sensors-19-00820-f001:**
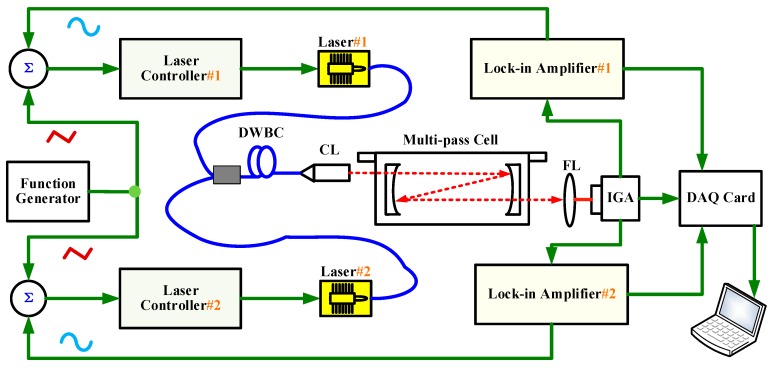
Schematic diagram of the dual-gas sensor. Laser #1 and #2: lasers for CH_4_/C_2_H_6_; DWBC: single mode standard coupler; CL: collimator; FL: focusing lens; IGA: InGaAs detector.

**Figure 2 sensors-19-00820-f002:**
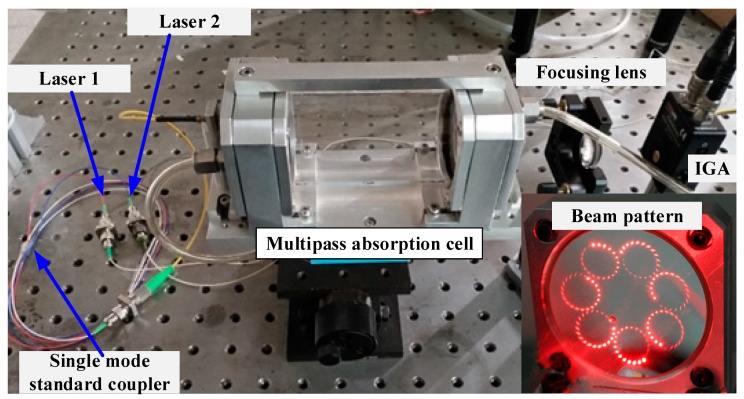
Photograph of the dual-gas sensor associated with a beam pattern (shown with a red laser) on one mirror of a home-made dense-pattern multipass cell.

**Figure 3 sensors-19-00820-f003:**
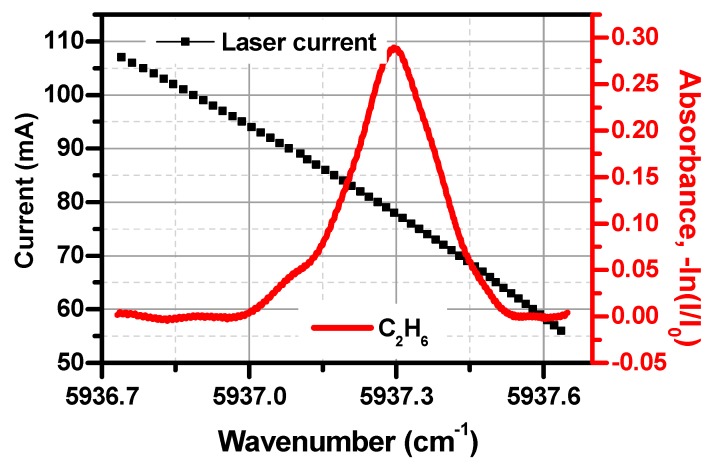
Absorption spectrum of ethane near 5937.3 cm^−1^. Left axis: laser current (vs. laser frequency); Right axis: absorbance of ethane in the tuning wavelength range.

**Figure 4 sensors-19-00820-f004:**
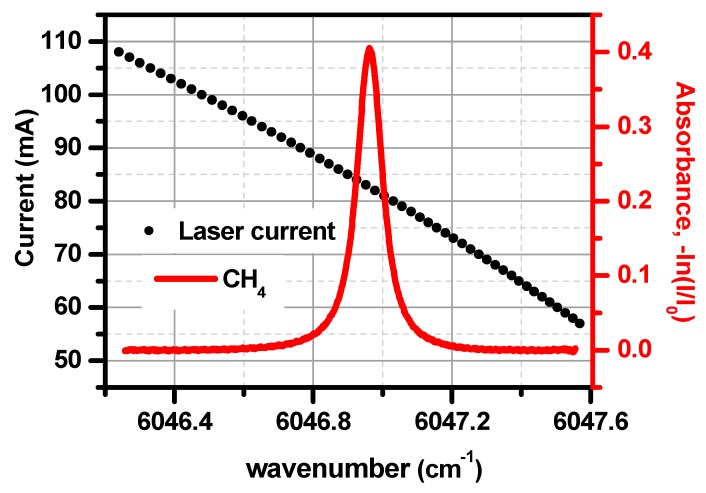
Absorption spectrum of methane near 6046.95 cm^−1^. Left axis: laser current (vs. laser frequency); Right axis: absorbance of the methane absorption line.

**Figure 5 sensors-19-00820-f005:**
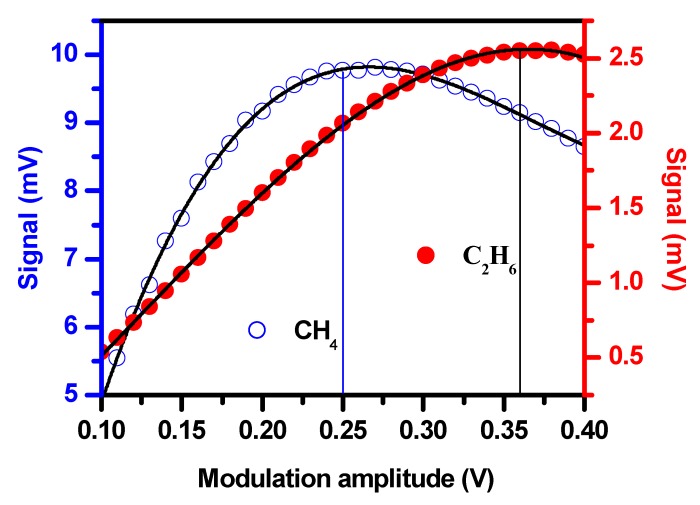
The 2f signals of CH_4_ and C_2_H_6_ vs. the modulation voltage amplitude.

**Figure 6 sensors-19-00820-f006:**
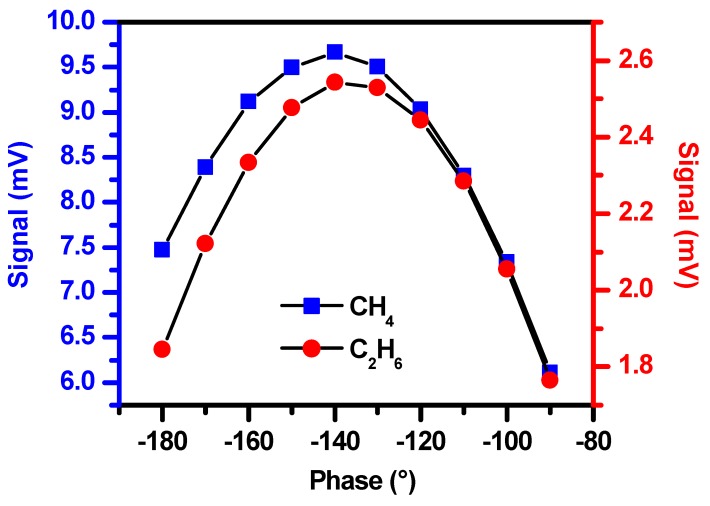
The 2f signal amplitudes of CH_4_ and C_2_H_6_ vs. modulation phase.

**Figure 7 sensors-19-00820-f007:**
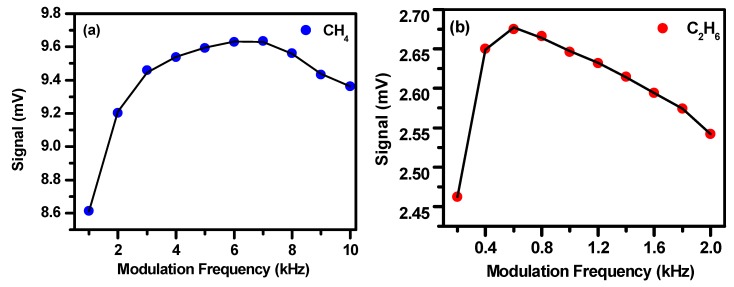
The 2f signal amplitudes vs. modulation frequencies for CH_4_ (**a**) and C_2_H_6_ (**b**).

**Figure 8 sensors-19-00820-f008:**
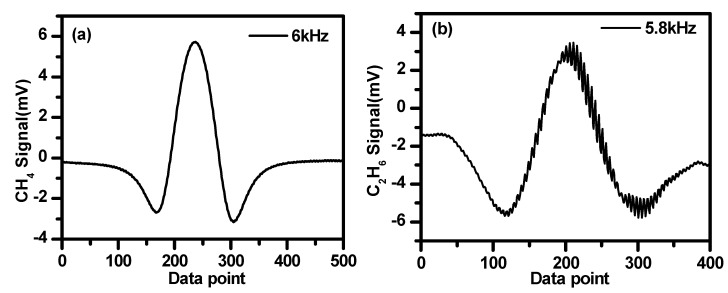
Second harmonic signals of CH_4_ modulated at 6 kHz (**a**) and C_2_H_6_ modulated at 5.8 kHz (**b**) showing cross-talk induced interference fringes superimposed on the second harmonic signal.

**Figure 9 sensors-19-00820-f009:**
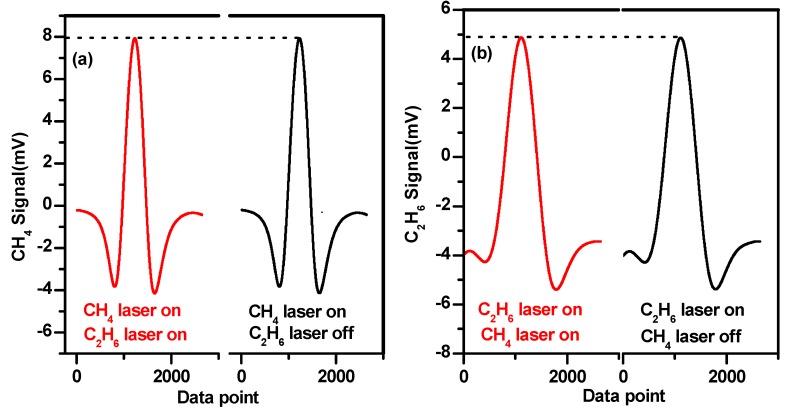
Comparison of the second harmonic signal amplitudes with two lasers on and only one laser on. (**a**) Methane detection and (**b**) ethane detection.

**Figure 10 sensors-19-00820-f010:**
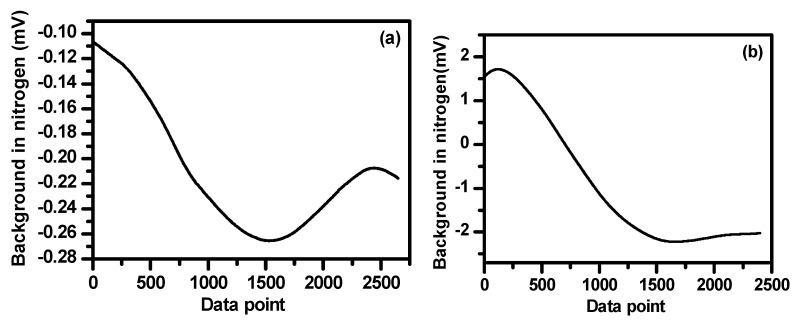
Baselines of nitrogen measured in the methane (**a**) and ethane (**b**) channels.

**Figure 11 sensors-19-00820-f011:**
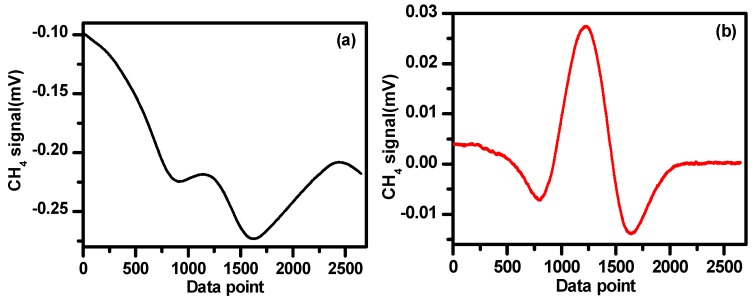
(**a**) 2f signal of 0.6 ppmv CH_4_ associated with baseline and (**b**) 2f signal of 0.6 ppmv CH_4_ after subtracting the baseline.

**Figure 12 sensors-19-00820-f012:**
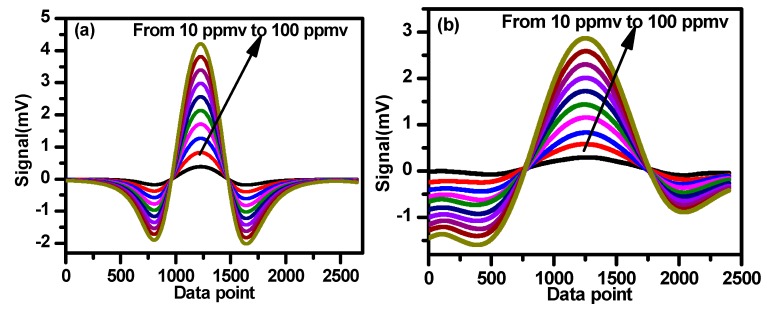
Second harmonic signals of CH_4_ (**a**) and C_2_H_6_ (**b**) at different concentrations ranging from 10 ppmv to 100 ppmv.

**Figure 13 sensors-19-00820-f013:**
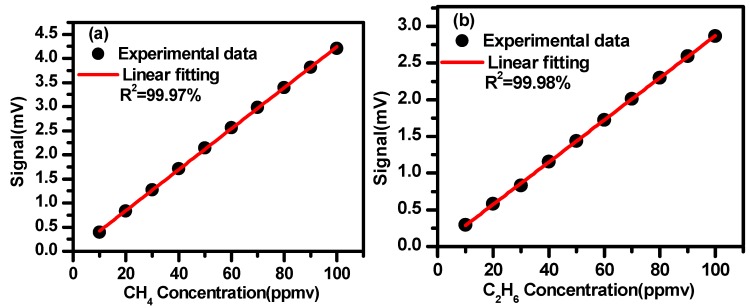
2f signal amplitudes vs. gas concentrations of CH_4_ (**a**) and C_2_H_6_ (**b**).

**Figure 14 sensors-19-00820-f014:**
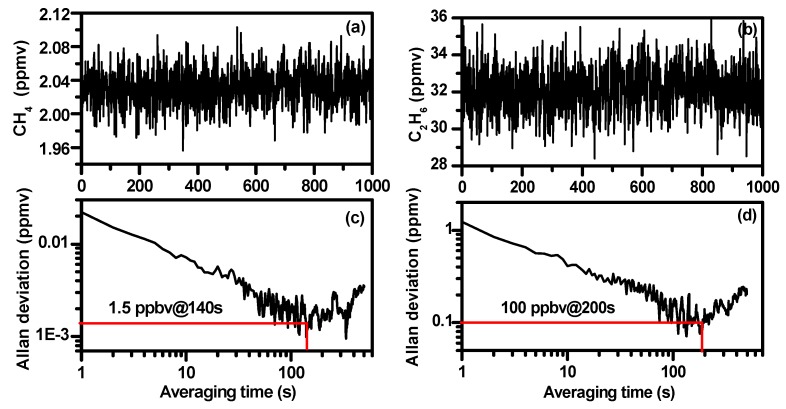
Time series measurements of CH_4_ (**a**) and C_2_H_6_ (**b**). Allan deviation plots for the CH_4_ (**c**) and the C_2_H_6_ (**d**) measurements.

**Figure 15 sensors-19-00820-f015:**
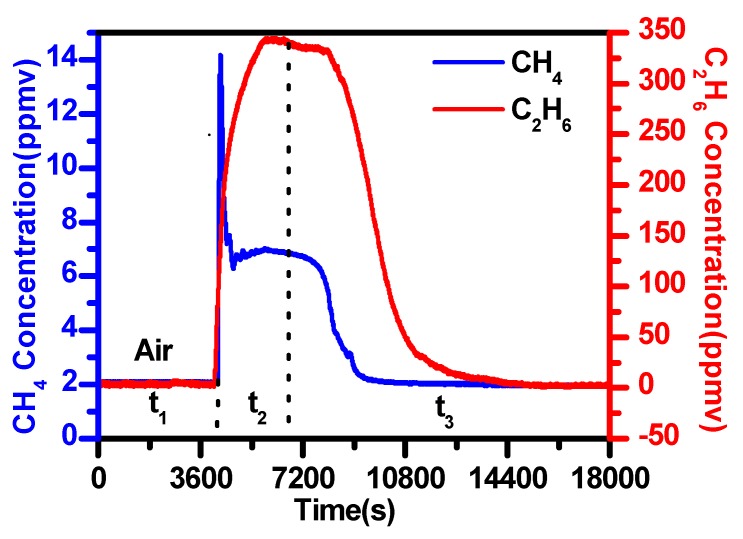
Simultaneous detection of methane and ethane “leakage”.

**Table 1 sensors-19-00820-t001:** Optimal modulation amplitude, phase, and frequency of CH_4_ and C_2_H_6_.

Target Gas	Optimal Modulation Amplitude	Optimal Modulation Phase	Optimal Modulation Frequency
CH_4_	0.25 V	−140°	6 kHz
C_2_H_6_	0.36 V	−140°	0.6 kHz

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
