# Peer review of "Dual-Gas Sensor of CH4/C2H6 Based on Wavelength Modulation Spectroscopy Coupled to a Home-Made Compact Dense-Pattern Multipass Cell"

_sensors, 2019, doi:10.3390/s19040820_

Round 1
Reviewer 1 Report
This Manuscript reporting on a Dual-gas sensor of CH4 / C2H6 exploiting a home-made compact dense-pattern multipass cell, sets the focus on one of the pivotal research and technological issues in oil exploration and natural gas analysis: the real-time and in-situ monitoring of the most important hydrocarbon molecules as indicators for natural gas characterization.
The motivations behind this work are solid, as well as the presentation of experimental results. The simultaneous detection of two gases by means of two lock-in amplifiers is a scheme already employed also in quartz-enhanced photoacoustic spectroscopy (“Simultaneous dual-gas QEPAS detection based on a fundamental and overtone combined vibration of quartz tuning fork” by H. Wu et al.) and the C2H6 absorption line at 1684 nm was also targeted for standard photoacoustic detection by G. Cheng et al. in “Photoacoustic measurement of ethane with near infrared DFB diode laser”. These two papers along with “Methane, ethane and propane detection using a compact quartz enhanced photoacoustic sensor and a single interband cascade laser” by A. Sampaolo et al. should be included into the reference list as representative of hydrocarbon detection via photoacoustic techniques.
Furthermore, before considering this manuscript for publication, some minor revisions must be addressed:
- The sentence from line 53 to 56 should be re-arranged: it is not clear which subject “with higher performances” is referred to.
- It is not clear why should be necessary to dedicate a paragraph to the well known wavelength modulation spectroscopic approach. The all derivation of the second harmonic term seems to be functional to the final take home message that the harmonic signal is proportional to the concentration of the gas – obvious – while the real deal here is the non-interfering 2f demodulation of the CH4 and C2H6 signals.
- More info should be provided about the 2x1 fiber combiner, the model info and the losses. The same for the collimator used before the injection into the MPC.
- Even if not necessary for the comparison, the concentration of methane and ethane for the signal shown in fig 9 should be specified anyway.
- It is not clear how the CH4-2 ppm and C2H6-32ppm concentrations used for the allan deviation were achieved. If these concentrations were obtained by diluting the 492ppm:N2 and 500ppm:N2 standard reference gas, this means that the dilution ratios was 1/250 for methane! In these case some words should be spent about the uncertainties on concentrations.
- The Allan deviation paragraph should be placed after the sensor calibration, otherwise the mV-ppm conversion in the Allan plots of fig 10 in ppm would not make sense.
- It is not clear if the background measurement in pure N2 for the baseline removal was made with the wavelength modulation on. Please specify it.
- It is not clear from lines 222, 224 if the baseline was substracted from the signal or the other way around.
- In fig. 11, the units are missing (still mV I guess).
- It should be specified that the measurements in fig. 15 were carried out with the current locked at the absorption peak.
- Line 283: the font is partially different.
Author Response
Dear reviewer,
We would like to thank you for your time and your constructive comments which allow us to improve the quality of this manuscript about our Dual-gas sensor. After thorough consideration, we have carefully addressed the issues raised by you.The point-by-point responses (in red) to the reviewers' comments are given below.

Reviewer 2 Report
This paper presents an interesting work about the dual-gas sensor measuring CH4 and C2H6. I would suggest a major revision of the manuscript, the reasons are listed as below:
1. Theory part: The authors present the basic theory of the second harmonic in wavelength modulation spectroscopy (WMS), which is not new and people have already done it many years ago. However, in the experimental results and discussions, the authors talk a lot about the modulation amplitude, frequency and phase, the potential cross-talk, and the baseline, I would suggest the authors present these parameters in the theory and then discuss the relationships between the second harmonic and these parameters.
2. In my opinion, Section 4.2 is unnecessary, because the phases would vary in different operations. The authors do not consider the phases which exit in the WMS correctly. For reference, I would suggest reading the Appendix I in Z. Peng’s article: Peng et al. (2013) -- ``First harmonic with wavelength modulation spectroscopy to measure integrated absorbance under low absorption".
3. I am wondering how to obtain the results in Fig. 7, because theoretically, the modulation frequency will not change the amplitude of the harmonic, when the cut-off frequency in the lock-in amplifier is set appropriately. However, I also used the lock-in amplifier before; it is hard to set the cut-off frequency in the SR 830 DSP in a very good statue to obtain the entire harmonic signal. The author should provide more information about the cut-off frequency and the signal gain in the lock-in amplifier, and also explain the results in Fig. 7.
4. Can the authors provide more information and explanation about the cross-talk effect?
5. "The calibration curves of laser current vs. frequency were realized with the help of a
140 wavelength meter (Bristol 621)", which is probably done only once at the beginging. How are you going to calibrate the sensor exactly during the measurements? Standard approaches include using etalon, seperate reference cell or inline reference cell, please see: Chen et al. (2011), “VCSEL-based Calibration-free Carbon Monoxide Sensor at 2.3 μm with in-line Reference Cell”. Please provide a discussion of the possible approaches and an explaination how you are going to calibrate the sensor for day-to-day measurements in the field.
6 Section 4.1 There are some papers discussing the relationship between modulation index m, i.e. wavelength modulation amplitude divided by linewidth, and the signal amplitude (R. Arndt (1965) "Analytical Line Shapes for Lorentzian Signals Broadened by Modulation") as well as the shape of the second harmonics (J. Chen (2010) "Laser spectroscopic oxygen sensor using diffuse reflector based optical cell and advanced signal processing"). Please add "m" values of your modulation index (not also in voltage) and provide a discussion why you are not using the recommended value of m = 2.2.
7. A detection limit of 100 ppbV is not sufficient for detection of natural gas leakage, please see McKain et al. (2015), "Methane emissions from natural gas infrastructure and use in the urban region of Boston, Massachusetts". Could you please explain what applications do you have in mind?
8. Can you explain a little about the baseline in Fig. 11? It seems that the fluctuations in the baselines are the absorbance. Can you explain it in theoretical formula derivations? And how are the units in the Y-axis in Fig. 11? Why the baseline in C2H6 is much stronger than that in CH4?
9. In Fig. 3, the C2H6 absorption line is overlapped by a weaker absorption line, do you consider the overlapping effect in the harmonic detection? How do you evaluate it?
Author Response

(The authors gave the same response as above.)
